# The Effect of Gum Arabic (*Acacia senegal*) on Cardiovascular Risk Factors and Gastrointestinal Symptoms in Adults at Risk of Metabolic Syndrome: A Randomized Clinical Trial

**DOI:** 10.3390/nu13010194

**Published:** 2021-01-09

**Authors:** Amjad H. Jarrar, Lily Stojanovska, Vasso Apostolopoulos, Jack Feehan, Mo’ath F. Bataineh, Leila Cheikh Ismail, Ayesha S. Al Dhaheri

**Affiliations:** 1Nutrition and Health Department, College of Medicine and Health Sciences, United Arab Emirates University, Al Ain 15551, UAE; amjadj@uaeu.ac.ae (A.H.J.); lily.stojanovska@uaeu.ac.ae (L.S.); 2Institute for Health and Sport, Victoria University, Melbourne 3011, Australia; vasso.apostolopoulos@vu.edu.au (V.A.); jfeehan@student.unimelb.edu.au (J.F.); 3Department of Medicine–Western Health, The University of Melbourne, Melbourne 3010, Australia; 4Department of Sport and Rehabilitation, The Hashemite University, 13115 Zarqa, Jordan; mfmbataineh@hu.edu.jo; 5Clinical Nutrition and Dietetics Department, College of Health Sciences, University of Sharjah, Sharjah 27272, UAE; lcheikhismail@sharjah.ac.ae; 6Nuffield Department of Women’s & Reproductive Health, University of Oxford, Oxford OX1 2JD, UK

**Keywords:** metabolic syndrome, Gum Arabic, *Acacia senegal*, *Acacia seyal*

## Abstract

Gum Arabic (GA) is a widely-used additive in food processing, but is also historically used in a number of traditional therapies. It has been shown to have a broad range of health benefits, particularly in improving important cardiovascular risk indicators. Metabolic syndrome and its associated cardiac outcomes are a significant burden on modern healthcare systems, and complementary interventions to aid in its management are required. We aimed to examine the effect of GA on those with, or at risk of, metabolic syndrome to identify an effect on improving important disease parameters related to cardiovascular outcomes. A single-blind, randomized, placebo-controlled trial was conducted to identify the effects of daily GA supplementation on metabolic and cardiovascular risk factors. A total of 80 participants were randomized to receive 20 g of GA daily (*n* = 40) or placebo (1 g pectin, *n* = 40) for 12 weeks. Key endpoints included body-anthropometric indices, diet and physical activity assessment, and blood chemistry (HbA1c, fasting glucose, and blood lipids). Of the 80 enrolled, 61 completed the study (intervention: 31, control: 30) with 19 dropping out due to poor treatment compliance. After 12 weeks, the participants receiving the GA showed significant decreases in systolic and diastolic blood pressure, fat-free body mass, energy and carbohydrate consumption, and fasting plasma glucose, as well as increased intake of dietary fiber. They also reported improvements in self-perceived bloating and quality of bowel movements, as well as a decreased appetite score following GA consumption. These results suggest that GA could be a safe and beneficial adjunct to other treatments for those with, or at risk of, metabolic syndrome.

## 1. Introduction

Gum Arabic (GA) or *Acacia* gum is a soluble dietary fiber obtained from the stems and branches of the *Acacia senegal* and *Acacia seyal* plants, which grow mainly in the African region of Sahe in Sudan [1]. It is often associated with health benefits relevant to reducing the risk of metabolic syndrome (MetS). GA contains three different fractions of highly-branched carbohydrate structures that vary in molecular mass and protein content, which are believed to underlie its physiological effects. These are commonly known as the arabinogalactan-protein, arabinogalactan, and glycoprotein fractions [2]. However, the composition of GA may change depending on the source, climate, and soil [3]. Because of the physical properties of GA, it has been widely used in various industries including cosmetics, textiles, ceramics, pharmaceuticals and foods [4]. GA is commonly used in industrial food production as an emulsifier, a stabilizer, and a thickener due to its nondigestibility, low-solution viscosity, and safety [5]. Used as a traditional remedy for many years, several studies have described the antioxidant properties of GA and its capacity to neutralize reactive oxygen substances [6,7,8]. Research also suggests it may have an effect on lipid metabolism [9], as well as renal function [10,11,12] and satiety [13], lending support to its use as an adjunct in the prevention and treatment of metabolic syndrome. Broadly, GA appears to have a hypocholesterolemic effect, decreasing low-density lipoproteins (LDL) and very low density lipoproteins (VLDL) without affecting high-density lipoproteins (HDL) and triglycerides in animal models [6]. GA has shown potential to relieve the effects of chronic renal failure by improving creatinine clearance as well as excretion of magnesium and calcium [14]. It has also been reported to decrease blood pressure in mice, and has been shown to lower caloric intake significantly, potentially due to increased dietary-fiber intake increasing satiety [5]. This reduction in energy intake makes GA a strong candidate for adjunct weight-control therapies.

Metabolic syndrome describes a cluster of conditions including increased blood pressure, high blood glucose, excess body fat, and dyslipidemia occurring simultaneously [15]. Metabolic syndrome is one of the most significant risk factors for a wide range of noncommunicable diseases (NCDs) such as cardiovascular diseases and diabetes [16]. According to the World Health Organization (WHO), NCDs are responsible for 71 percent of deaths globally, with cardiovascular disease being the leading cause of death, followed by cancer, respiratory diseases, and diabetes [17]. Most NCDs share common behavioral risk factors such as poor diet, physical inactivity, and smoking, as well as the key MetS risk factors such as overweight, obesity, high blood sugar, and hypercholesterolemia [18]. This makes reducing the burden of MetS a key element in the prevention of NCDs [16].

Given the significant burden of MetS and its associated risk of NCDs, there is a need for novel interventions to help prevent its onset. This study aimed to investigate the effect of consuming 20 g of Gum Arabic-Acacia Senegal (GA-AS) per day based on key metabolic parameters in adults with, or at risk of, metabolic syndrome. It is the hope that the results of this study will provide insight into the dietary effects of consistent consumption of Gum Arabic-Acacia Senegal. It is hypothesized that GA-AS will provide benefits to the metabolic health of the participants with regular use and will be well tolerated by individuals taking it.

## 2. Materials and Methods

### 2.1. Study Design

The present study was a controlled, randomized, single-blind, parallel-design study comparing an intervention group receiving 20 g of GA-AS daily for 12 weeks with a control group receiving a daily placebo containing 1 g of pectin for the same period. The primary endpoints of the study were blood glucose, lipid profile, blood pressure, body composition, gastrointestinal motion, and satiety. The study was conducted at the department of Nutrition and Health, College of Food and Agriculture at United Arab Emirates University (UAEU) during the period from January to May 2018. This study was conducted according to the principles of the Helsinki declaration on human research ethics and was approved by the UAEU scientific research ethics committee (ref. no. ERH_2016_4372)**.**

### 2.2. Study Participants

Participants were recruited from the UAEU (students and staff) through face-to-face interviews, email, social media, and printed advertisements on the campus and in the dormitories of the university. Participants were given both verbal and written information about the aim of the study, data to be collected, and the duration of intervention, and they were required to sign a written informed-consent form to participate in the study prior to screening.

Participants were screened for eligibility at the nutrition clinic at UAEU. Participant eligibility was based on the presence of metabolic-syndrome risk factors. Inclusion and exclusion criteria are summarized in Table 1. Risk factors assessed included waist circumference (females > 80 cm and males > 94 cm), systolic blood pressure (≥130 mm Hg), diastolic blood pressure (≥80 mm Hg), blood-fasting glucose (≥100 mg/dL), HDL cholesterol level (for female < 50 mg/dL and males < 40 mg/dL), and triglycerides level (≥150 mg/dL). Participants with more than three risk factors, or with two risk factors and one borderline, were included in the study. Participants were excluded from the study if they were smokers, pregnant women, lactating women, or were taking permanent medication.

Participants were asked to complete a health-screening questionnaire that contained questions about medical conditions and medications that might influence glucose control, appetite, and energy expenditure. All participants signed an informed-consent form before taking part in the study. This study was conducted according to the guidelines in the Declaration of Helsinki. All procedures involving human subjects were approved by the United Arab Emirates University (UAEU) Scientific Research Ethics Committee.

Participants were randomly assigned to control and intervention groups via computer software, with the experimental group receiving 20 g of GA-AS powder per day and the control group receiving 1 g of placebo (pectin) powder per day for a period of 12 weeks. This dose was selected based on previous research showing metabolic effects with doses of 10–30 g per day for 4–12 weeks, with lipid effects being most significant after 5 weeks [3,13,19,20]. GA powder and placebo were provided in premeasured sachets, and participants were asked to consume the GA powder or the placebo two times per day by adding it to hot water, tea, milk, or on any meal. The nutrient composition of the study dose of Gum Arabic is presented in Table 1. Body weight (kg), height (cm), waist circumference (cm), body composition, blood glucose, blood-lipid profile, glycated hemoglobin A1c (HbA1c), and blood pressure were measured at baseline (week 0) and 12 weeks. In addition, participants were asked to complete a bowel-movement questionnaire and satiety scale at the baseline and endpoint of the study.

### 2.3. Research Parameters

#### 2.3.1. Anthropometric Measurements

Body weight was recorded to the nearest 0.01 kg while the subject was wearing minimal clothes (as per local cultural requirements) and no shoes. Body composition was assessed via a bioimpedance device (InBody720, InBody, CA, USA), providing measurement of percentage body fat (%BF), fat mass (kg), and fat free mass (kg). Waist circumference was measured using measuring tapes, according to standard methods at the mid-point between inferior costal margin and superior border of the iliac crest. In obese individuals, the measurement was taken at the level of the umbilicus [21]. Body mass index (BMI) was calculated as BMI = kg/m^2^. All measurements were taken at baseline and after 12 weeks of intervention.

#### 2.3.2. Diet and Physical Activity Assessment

During the study period, the participants were asked to maintain their normal lifestyle. Participants were asked to record their dietary intake at baseline (week 0) and at week 12 of the study period. Food records were taken over three days including two weekdays and one weekend day. Photographs of food with different portion sizes were used to help participants estimate the correct portion size consumed. The Food Processor^®^ Nutrition and Fitness Software, ESHA food-analysis program (version 10.4), and the Kuwaiti Food Composition database were used to analyze the energy and nutrient contents of the consumed foods [22].

Physical activity level was assessed using the International Physical Activity Questionnaire (IPAQ) (Arabic and English versions) at baseline and week 12 [23].

#### 2.3.3. Bowel Movement and Satiety Questionnaires

Participants were asked to answer a bowel-movement questionnaire including frequency and intensity of constipation, bloating, diarrhea, and heartburn. A satiety questionnaire with scoring points was used to assess satiety after 60 min of ingestion of either GA-AS or the placebo intervention. The questionnaires were administered at both baseline and at the end of the study (adapted from [24,25]). In the satiety questionnaire, participants answered the following questions: How did the meal (with the study treatment or placebo) you just ate make you feel? Did it satisfy your hunger, or did you feel like you needed to snack later? Then they rated their feelings of satiety for 60 min using a score of 100 [24,25].

#### 2.3.4. Biochemical Parameters

A fasting, venous blood sample was collected (5 mL) by a certified phlebotomist at baseline and at end of the study period. Fasting blood glucose (FBG), HbA1c, serum triglyceride (TG), high-density lipoprotein cholesterol (HDLC), low-density lipoprotein cholesterol (LDLC), and total cholesterol (TC) were analyzed using Cobas C111 automated biochemical analyzer (Roche Diagnostics, Indianapolis, IN, USA). All data were collected at the laboratory facilities of the Nutrition and Health Department.

### 2.4. Statistical Analysis

G*Power 3.1.9.2 software was used for sample-size calculation of repeated measures ANOVA with parallel design. Power calculation identified a sample size of at least 54 participants to detect a medium-effect size (0.25) with 95% power with significance level set at 0.05. The statistical analysis was performed using SPSS version 24.0 and results presented as (Mean ± Standard Deviation). Repeated measures ANOVA was used to detect the main effects of time and group on study measures. Paired t-tests and independent t-tests were employed to compare the effect of time and groups (Control vs. Intervention), respectively. Binary data was assessed for statistical significance with the N-1 chi square test. Results were considered statistically significant at *p*-value < 0.05.

## 3. Results

The study sample consisted of 80 participants from the United Arab Emirates University, aged 18–50 years with a mean age of 25.51 ± 9.5 years, mean BMI of 33.9 ± 5.4, and with 62.3% being female. Nineteen participants dropped out of the study due to poor treatment compliance, with 61 ultimately completing the 12-week intervention. Dropouts were largely due to failure to take prescribed GA or placebo, or failure to present for follow up testing. The control group consisted of 30 participants, and the experimental group 31 (Figure 1). There were no significant differences in baseline characteristics between groups (Table 2).

After 12 weeks of GA treatment, the experimental group showed significant decreases in both systolic (*p* = 0.008) and diastolic blood pressure (0.009), as well as fat free mass (*p* = 0.03), with no intragroup difference in the control group. No significant inter- or intragroup differences were observed in BMI, waist circumference, or body fat between baseline and week 12 (Table 3).

### 3.1. Diet and Physical-Activity Assessment

After 12 weeks of GA treatment, the intervention group showed a decrease in carbohydrate (*p* = 0.008) and calorie (*p* = 0.014) intake and an increase in dietary fiber consumption (*p* ≤ 0.001), with no intergroup change in the controls (Table 4). There were also intergroup differences in carbohydrate (*p* = 0.004) and dietary-fiber (*p* ≤ 0.001) consumption at 12 weeks. The intervention group also showed a trend toward lower energy intake compared to controls after 12 weeks; however, this did not reach significance (*p* = 0.069). There were no inter- or intra-group changes to physical activity parameters either in fat or protein consumption (Table 4).

### 3.2. Biochemical Assessment

After 12 weeks of GA treatment, the intervention group had significantly decreased in fasting plasma glucose (*p* = 0.046), with the control group showing no change; however, there was no difference between the groups at the endpoint (*p* = 0.101). There were no changes to HbA1c or blood lipid profile in either group (Table 5).

### 3.3. Bowel Movement

The intervention group reported significant reductions (*p* = 0.005) in bloating and improvement in bowel movement (0.047) compared to the control group. Although there were no statistically significant differences in abdominal pain, better digestion, or reduction in nausea, the intervention rate responses were almost double those of the control group (Table 6).

### 3.4. Feeling of Satiety

One hour after taking the GA treatment, the participants in the intervention group showed a significant increase in appetite score (reflective of a decreased appetite) compared to controls receiving the placebo treatment (*p* = 0.01) (Table 6).

## 4. Discussion

To the best of our knowledge, the current study is the first to examine the effect of Gum Arabic on individuals with, or at risk of developing, metabolic syndrome. The daily treatment with GA caused improvement in a number of parameters important to modifying outcomes and risk of metabolic syndrome, including fasting plasma glucose, blood pressure, and energy intake. However, contrary to some other studies, our trial found no significant reduction in body weight or BMI. In one study, 30 g of GA per day for 6 weeks caused significant reductions in BMI, body-fat percentage, and weight in adult females compared to placebo-treated controls [3]. Another study on daily GA administration (30 g) for three months also found that BMI, visceral-adiposity index, and body-adiposity index were significantly lowered in the intervention group [26]. The difference in findings may reflect the relative size of the studies, as, while in our study these parameters did not reach significance, there was some evidence of a trend. With a larger group, the results may have become significant. The differences may also be due to differences in patient demographic. In the first study, the participants were younger and generally healthier, and while in the second study they were significantly older, they had a much lower BMI.

The demonstrated decreases in energy and carbohydrate consumption are significant in the context of metabolic syndrome, characterized in large part by impaired glucose metabolism and obesity. Reducing the intake of these through nonpharmacological supplementation could provide a means of management or prevention of the syndrome. While the exact mechanism underlying the behavior change is unclear, it may be due to the increased feelings of satiety following GA administration. These findings are echoed in another study, which showed that supplementation with two blends of GA (EmulGold1 (EG) and PreVitae1 (PV)) decreased the caloric intake significantly three hours after consumption and increased subjective ratings of satiety [19]. An increase in fat free mass (FFM) was also identified in this study. While interesting and potentially of benefit to the population studied, these results should be interpreted with caution. The bioimpedance device used in this study reports two-compartment FFM which measures bone, muscle, connective tissue, and water as one. While this is a reliable measure of body fat, it is unable to discriminate between beneficial changes in muscle or bone mass from increases in water retention [27]. To evaluate a potential change in bone or muscle parameters, measuring body composition by either four-compartment or dual-Xray absorptiometry (DXA) is required. This provides some support for the clinical use of GA in the management of obesity, potentially providing a means to lower the burden of invasive bariatric surgery or harsh pharmacotherapeutic avenues.

Another significant finding of the study at hand was its significant impact on the blood pressure of the participants receiving GA. As the most significant outcomes of metabolic syndrome are cardiovascular diseases, such as myocardial infarction and stroke, interventions lowering cardiac risk are particularly important. Again, the specific mechanism for this is unclear; however, it has been reported that intake of dietary fiber, including GA, was associated with a significant fall in mean systolic blood pressure [SBP] in normal individuals who neither had hypertension nor diabetes [10]. Another study of GA treatment in people living with diabetes also found a decrease in blood pressure with SBP decreasing by 5.9% and diastolic blood pressure (DBP) by 5.4% [26].

Gum Arabic is a soluble fermentable fiber that has shown hypoglycemic, antioxidant effects and improved lipid metabolism in previous studies [6,26,28]. Our data showed a significant reduction in blood glucose in the intervention group, a key parameter of metabolic syndrome. An animal model showed that GA has a glucose-lowering effect in rabbits with alloxan–induced diabetes. They showed that GA (at doses of 2, 3, and 4 mg/kg) significantly reduced the blood-glucose concentration of normal but not diabetic rabbits. They therefore concluded that GA initiated the release of insulin from pancreatic b cells in normal rabbits [29]. Another study suggested that the glycemic effects of GA may be due to its viscosity, which slows gastric emptying and alters the absorption kinetics in the intestine [30]. It has also been suggested that GA may have a prebiotic effect, which may underlie some of its metabolic effects [20], as it has recently been identified as a novel modulator of lipid profiles in vivo. Studies of GA have also demonstrated that consumption of 10 g/d for four weeks is associated with higher numbers of bifidobacteria and lactobacilli [19], both of which have been associated with beneficial effects on health in vivo.

In this study, 51% of the participants in the intervention group reported a significant improvement in reducing bloated feelings after 12 weeks of intake of GA-AS. The intervention group reported better responses in bowel movement, reduction in abdominal pain, better digestion, and reduction in nausea, but the results were insignificant compared with the control group. These findings may be due to the high dietary-fiber content in GA-AS (85%), which aids in healthy digestion and bowel movement. In addition to what has been previously reported, GA is not degraded in the stomach and small intestine, but undergoes complete fermentation within the cecum of rats [31,32] and humans [33]. Such fermentation promotes bacterial proliferation, which contribute to the prebiotic effect of GA [13,34].

## 5. Conclusions

Daily ingestion of 20 g of GA-AS for 12 weeks was shown to improve satiety and significantly reduce energy and carbohydrate intakes. It also improves blood pressure, blood glucose, and bowel movement, while increasing perceived satiety. This positions GA as a potential addition to the management of those with or at risk of developing metabolic syndrome. GA is a strong candidate for supplementation, as it is edible, safe, and already widely available in industry settings. It appears to have beneficial effects in a number of key areas relevant to improving the outcomes of those with metabolic syndrome, and could be a means of lowering the disease burden of NCDs globally. Future large-scale trials should evaluate the long-term use of GA in patients with metabolic syndrome to clarify its effects as well as identify optimal dose strategies and long-term efficacy.

## Figures and Tables

**Figure 1 nutrients-13-00194-f001:**
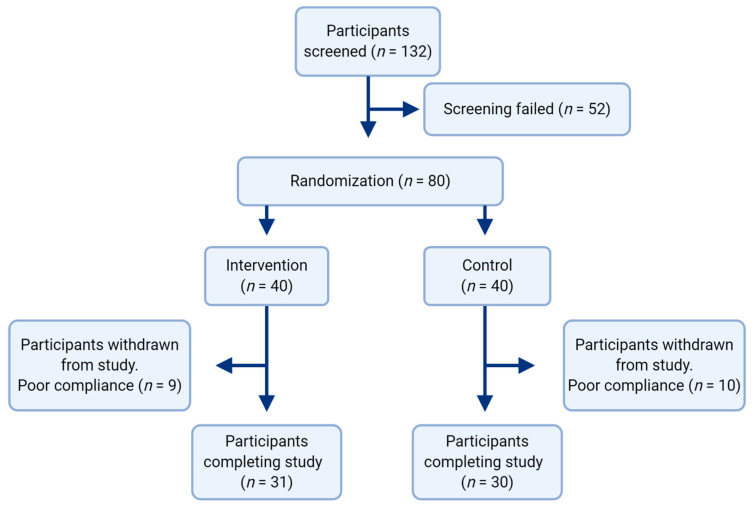
Summary of study recruitment.

**Table 1 nutrients-13-00194-t001:** Nutrient composition of GA per 20 g dose.

Nutrient	Per 20 g Dose
Energy (kcal)	1.8
Protein (g)	0.4
Carbohydrate (g)	17.1
Fat (g)	0.02
Total dietary fiber (g)	17.1
Sodium (mg)	2.8
Calcium (mg)	214.8
Magnesium (mg)	78.0
Potassium (mg)	182.8
Iron (mg)	0.2

**Table 2 nutrients-13-00194-t002:** Baseline characteristics of the subjects (*N* = 61).

Variable	Control (*n* = 30) Mean ± SD	Intervention (*n* = 31) Mean ± SD	*p*-Value
Age (years)	25.6 ± 9.9	28.3 ± 11.8	0.452
Weight (kg)	91.7 ± 20.8	92.1 ± 17.4	0.948
Height (cm)	168.3 ± 11.6	164.3 ± 7.5	0.163
BMI (kg/m^2^)	31.9 ± 4.7	34.1 ± 5.9	0.174
Waist circumference (cm)	100.5 ± 16.1	101.2 ± 12.7	0.867
Body Fat (%)	39.7 ± 8.4	43.7 ± 7.6	0.088
Fat free mass (kg)	53.8 ± 14.5	50.9 ± 9.4	0.382
Systolic Blood Pressure (mm Hg)	114.8 ± 16.4	118.3 ± 17.0	0.482
Diastolic Blood Pressure (mm Hg)	75.8 ± 9.9	81.1 ± 9.2	0.066
Physical Activity Levels			
Vigorous (min/week)	11.1 ± 3.0	14.4 ± 3.2	0.683
Moderate (min/week)	59.4 ± 5.7	62.1 ± 8.9	0.117
Light (min/week)	190.2 ± 13.1	193.5 ± 3.2	0.939
Sedentary Activity (h/day)	9.20 ± 0.4	10.5 ± 0.35	0.155
Nutritional Intake			
Energy (kcal)	2142 ± 551.8	2036.9 ± 601.5	0.534
Carbohydrate (g)	256 ± 57.7	239.4 ± 84.3	0.446
Fat (g)	82.2 ± 34.1	80.5 ± 30.7	0.852
Protein (g)	81.7 ± 28.9	86.9 ± 38.4	0.609
Dietary fiber (g)	15.0 ± 9.8	17.1 ± 15.2	0.648
Biochemical Parameters			
HbA1c (%)	6.1 ± 0.9	6.0 ± 1.7	0.763
Glucose (mg/dL)	101.5 ± 14.0	105.6 ± 36.0	0.635
Triglycerides (mg/dL)	94.7 ± 41.6	100.9 ± 53.9	0.661
Total Cholesterol (mg/dL)	150.7 ± 34.3	157.9 ± 28.2	0.413
LDLC (mg/dL)	2.5 ± 0.9	2.50 ± 0.7	0.926
HDLC (mg/dL)	46.3 ± 12.1	45.1 ± 12.1	0.747

**Table 3 nutrients-13-00194-t003:** Changes in physical characteristics for study population after 12 weeks.

	Control			Intervention			Intergroup Difference (Week 12)
Variable	Baseline (Mean ± SD)	Week 12 (Mean ± SD)	*p*-Value	Baseline (Mean ± SD)	Week 12 (Mean ± SD)	*p*-Value	*p*-Value
Weight (kg)	91.74 ± 20.8	93.0 ± 22.3	0.37	92.09 ± 17.4	91.43 ± 17.0	0.116	0.778
BMI (kg/m^2^)	31.92 ± 4.7	32.6 ± 5.6	0.288	34.07 ± 5.9	33.90 ± 6.0	0.465	0.43
Waist circ. (cm)	100.50 ± 16.1	100.6 ± 18.9	0.937	101.20 ± 12.7	99.07 ± 13.0	0.155	0.728
Body Fat (%)	39.72 ± 8.4	40.90 ± 8.90	0.206	43.70 ± 7.6	44.10 ± 7.8	0.094	0.198
Fat Free Mass (kg)	53.79 ± 14.5	58.80 ± 9.70	0.185	50.85 ± 9.4	55.38 ± 8.9	0.030 *	0.206
Systolic (mm Hg)	114.80 ± 16.4	117.0 ± 15.0	0.242	118.30 ± 17.0	111.30 ± 19.8 *	0.008 *	0.273
Diastolic (mm Hg)	75.8 ± 9.90	79.50 ± 9.7	0.102	81.1 ± 9.2	76.70 ± 13.2 *	0.009 *	0.419

* *p* = < 0.05.

**Table 4 nutrients-13-00194-t004:** Dietary and physical-activity characteristics of the study population.

	Control			Intervention			Intergroup Difference (Week 12)
Variable	Baseline (Mean ± SE ^1^)	Week 12 (Mean ± SE ^1^)	*p*-Value	Baseline (Mean ± SE ^1^)	Week 12 (Mean ± SE ^1^)	*p*-Value	*p*-Value
**Vigorous (min/week)**	11.1 ± 3.0	10.7 ± 2.8	0.101	14.4 ± 3.2	18.4 ± 6.0	0.91	0.55
**Moderate (min/week)**	59.4 ± 5.7	71.5 ± 16.6	0.525	62.1 ± 8.9	59.0 ± 10.2	0.665	0.709
**Light (min/week)**	190.2 ± 13.1	206.5 ± 14.2	0.408	193.5 ± 3.2	198.4 ± 20.6	0.923	0.863
**Sedentary Activity (h/day)**	9.20 ± 0.4	10.9 ± 0.4	0.089	10.5 ± 0.35	10.7 ± 0.43	0.677	0.847
**Energy (kcal)**	2142 ± 62.4	2092 ± 55.0	0.644	2036.9 ± 68.0	1810 ± 62.5 *	0.014	0.069
**Carbohydrate (g)**	256 ± 6.5	256 ± 6.7	0.976	239.4 ± 9.5	194.1 ± 8.8 *	0.008	0.004
**Fat (g)**	82.2 ± 3.9	73.2 ± 3.11	0.259	80.5 ± 3.5	71.3 ± 3.2	0.139	0.815
**Protein (g)**	81.7 ± 3.3	87.0 ± 3.6	0.453	86.9 ± 4.3	84.9 ± 5.2	0.836	0.854
**Dietary fiber (g)**	15.0 ± 1.1	16.1 ± 1.2	0.829	17.1 ± 1.7	31.9 ± 1.64 *	0.001	<0.001

^1^ Standard Error, * *p =* <0.05.

**Table 5 nutrients-13-00194-t005:** Biochemical measurements of study population.

	Control			Intervention			Intergroup Difference (Week 12)
Variable	Baseline (Mean ± SD)	Week 12 (Mean ± SD)	*p*-Value	Baseline (Mean ± SD)	Week 12 (Mean ± SD)	*p*-Value	*p*-Value
HbA1c (%)	6.10 ± 0.90	6.0 ± 0.4	0.662	6.0 ± 1.7	6.00 ± 0.8	0.938	0.145
Glucose (mg/dL)	101.50 ± 14.00	99.5 ± 14.2	0.489	105.6 ± 36.0	92.90 ± 13.20 *	0.046	0.55
Triglycerides (mg/dL)	94.70 ± 41.60	94.9 ± 41.0	0.955	100.9 ± 53.9	93.90 ± 44.0	0.936	0.832
Total Cholesterol (mg/dL)	150.7 ± 34.30	151.2 ± 37.8	0.941	157.9 ± 28.20	152.60 ± 30.6	0.892	0.709
LDLC (mg/dL)	2.50 ± 0.90	2.3 ± 0.8	0.433	2.50 ± 0.70	2.4 ± 0.7	0.625	0.925
HDLC (mg/dL)	46.3 ± 12.1	46.2 ± 13.2	0.951	45.1 ± 12.1	44.8 ± 13.9	0.736	<0.001

* *p =* <0.05.

**Table 6 nutrients-13-00194-t006:** The effect of GA on the bowel movement at week 12 in the Control and Intervention Groups.

Response Rate for	Week 12		
	Control (Yes, %)	Intervention (Yes, %)	*p*-Value
Improved bowel movements	33.3	54.8	0.047 *
Reduction in bloating feelings	20.0	51.6	0.005 *
Reduction in abdominal pain	10.0	22.6	0.094
Feeling of better digestion	23.3	41.9	0.062
Reduction in nausea	10.0	25.8	0.056
Satiety Scores	Baseline (Mean ±SD)	60 min (Mean ±SE)	
Control	44.9 ± 24.7	51.6 ± 24.2	0.174
Gum Arabic	48.7 ± 22.3	62.5 ± 27.5	0.011

* *p =* <0.05.

## Data Availability

The data presented in this study are available on request from the corresponding author. The data are not publicly available due to university privacy guidelines.

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
