# Peer review of "The Effect of Gum Arabic (*Acacia senegal*) on Cardiovascular Risk Factors and Gastrointestinal Symptoms in Adults at Risk of Metabolic Syndrome: A Randomized Clinical Trial"

_nutrients, 2021, doi:10.3390/nu13010194_

Round 1
Reviewer 1 Report
In this clinical trial, Jarrar and colleagues attempt to elucidate some of the health benefits of Gum Arabic. While the presentation of the data is appropriate. This paper lacks novel and in-depth metabolic analyses.
Minor points.
The introduction is far too long. There are several spacing errors that can be distracting. Line 70-84 seem superfluous.
Some minor spelling errors throughout.
Major points.
While there is an effect for lower fasting glucose the HbA1c data shows no change. Was plasma insulin measured? If there is a change in fasting glucose one would think HbA1c should be different over that duration of time.
The differences in fat-free mass are not explained in a sufficient manner. The layout of the results makes it confusing...especially table 1. This section could be reorganized.
Ultimately, there is a lack of novel biomarkers or a hypothesis-driven approach that limits the enthusiasm for this paper.
Author Response
We thank the reviewer for their feedback, please see the attached response document and revised manuscript.

Reviewer 2 Report
Dear Authors,
Pretty good background to the problem and explanation of the GA and MetS. Authors explore an interesting area of GA use. To verify introduction section properly I should have a chance to see the whole list of cited papers. I have found that references 31-42 are missing. Due to a mess in tables and figures numeration it is hard to follow the text. Authors should include an appropriate table 1 and 3 and revised the rest of them.
- It would be beneficial for a "discussion" section to explore prebiotic effect of GA in more details. May it help to explain better the study results?
- Could the Authors explain, why they decided to give 20 g of GA to participants?
- line 22 (abstact): A total of 80 participants were randomized to receive 20g of GA daily (n=40) or placebo (1g pectin, n=40) per day for 12 weeks...... and line 24 .....Of the 80 enrolled, 61 completed the study.........
but line 190 says: The study sample consisted of 61 participantsIn my opinion section "2.2. subjects" should contain more detailed description of study sample eg. the text that was included in the abstract. The Authors may also consider to change the title of section 2.2. eg. "study participant group"? - line 25:...." with 19 dropping out due to poor treatment compliance...."- What does it mean for Authors?
- Line 109: ..............inclusion and exclusion criteria are summarized in table 1...." I can see that table 1 is entitled: Nutrient composition for GA-AS per 10 g". Table "Inclusion and exclusion criteria" is missing.
- Line 115: as previous
- line 127- composition of GA is presented in table 1: it should be a table 2.
- line 162: PLLT5W8C- what does it mean?
- line 202 we can find another table 1
- line195 and 213 we can see two tables nb 2
- line 222 table 3 is missing and it wasn't mentioned in the manuscript
- line 228 table 6 -is missing
- line 234 figure 2 but we can just see figure 1 in the main text -line 236
Author Response
We thank the reviewer for their valuable feedback. Please see the attached response and revised manuscript

Reviewer 3 Report
Interesting paper. Large dose of fiber, but seems to be well tolerated.
Author Response
We thank the reviewer for their feedback. Please see the revised manuscript